# Assessing the impact of global warming on the distributions of *Allium stipitatum* and *Kelussia odoratissima* in the Central Zagros using a MaxEnt model

**Farzaneh Khajoei Nasab** [ID]*, **Amin Zeraatkar***

Research Division of Natural Resources, Chaharmahal and Bakhtiari Agricultural and Natural Resources Research and Education Center (AREEO), Shahrekord, Iran

\* a.zeraatkar@areeo.ac.ir (AZ); Farzaneh.khajoei@yahoo.com (FKN)

## Abstract

Global warming is an undeniable fact occurring in different parts of the world. Climate changes can have irreversible effects on plant communities, particularly on endemic and endangered species. Therefore, it is important to predict the impact of climate change on the distribution of these species to help protect them. This study utilized the MaxEnt model to forecast the impact of climate change on the distributions of two medicinal, edible, and aromatic species, *Kelussia odoratissima* and *Allium stipitatum*, in Chaharmahal and Bakhtiari province. The study used the CCSM4 general circulation model along with two climate scenarios, RCP2.6 and RCP8.5, for the 2050s and 2070s to predict the potential impact of climate change on the distribution of the species studied. The research findings indicated that the model performed effectively for prediction (AUC≥0.9). The primary environmental variables influencing species distribution were found to be isothermality (Bio3), soil organic carbon, and pH for *A. stipitatum*, and soil organic carbon, precipitation seasonality (Bio15), and precipitation of the wettest month (Bio13) for *K. odoratissima*. The findings suggest that the distribution of the studied species is expected to decline in the 2050s and 2070s due to climate change, under both the RCP2.6 and RCP8.5 climate scenarios. The research indicates that climate change is likely to have a significantly negative effect on the habitats of these species, leading to important ecological and socio-economic impacts. Therefore, our study emphasizes the urgent need for conservation efforts to prevent their extinction and protect their habitats.

## Introduction

Many endemic or rare species of medicinal plants are at risk of extinction [1,2,3]. The reasons for this phenomenon are varied, including limited distribution [4], declining population size [5], overexploitation [6], and low reproductive capacity [7]. It is disheartening to know that ecosystems rich in species, including plants important for medicinal and food purposes, are disappearing due to habitat destruction and unsustainable resource exploitation [8,9]. Climate change, driven by harmful human activities, leads to irregularities in temperature and rainfall

**Data availability statement:** Datasets analyzed during the current study are available on Figshare at https://doi.org/10.6084/m9.figshare.28557035.v1.

**Funding:** This research was done as part of the Post Doctorate Program of F. KHN. under the supervision of A.Z which was financially supported by Iran National Science Foundation: INSF (Grant No. 4020273).

**Competing interests:** The authors have declared that no competing interests exist.

patterns. This causes some alpine areas to experience greening due to heavy rainfall, while other regions suffer from drought [10,11]. These abnormal changes disrupt the ecosystem and contribute to the spread of plant diseases, pests, and parasites [12,13]. This not only leads to the loss of valuable species but also the loss of cultural diversity associated with them, occurring at an alarming rate [14]. It is crucial to implement effective monitoring and conservation efforts to protect these species.

The impact of climate change on plant communities is a crucial area of research. Understanding this impact helps scientists make informed decisions in preparation for future crises [15]. Species distribution models (SDMs) are commonly used numerical tools for mapping and predicting spatial distribution of species based on environmental factors [16,17,18]. These models require geolocated presence or presence-absence data, along with environmental variables [19]. The MaxEnt model is particularly popular among SDMs for accurately simulating the geographic distribution of living organisms [20,21,22,23]. It has demonstrated superior performance compared to other models, especially when dealing with limited sample sizes and presence-only data [24]. MaxEnt is extensively used for protecting vulnerable species [25], identifying management areas for invasive species [26], and assisting in pest and disease control [27].

In Iranian culture, plants and plant-based products hold significant role in both material and spiritual aspects [28]. Iranians rely on plants for various purposes and use around 2,075 plant species as herbal medicine [29]. However, the excessive use of these plants has led to the endangerment of several valuable plant species in Iran [30,31,32].

*Kelussia odoratissima* Mozaff. and *Allium stipitatum* Regel are two medicinal, edible, and aromatic plant species highly valued, particularly those residing in the Zagros region. The *Allium stipitatum* Regel, commonly known as the Iranian shallot, is an important member of the Amaryllidaceae family, which is naturally found in Iran (Fig 1a, b). This aromatic plant has been used as a spice and flavoring in Iranian cuisine for a long time and is commonly included in various types of pickles. The Iranian shallot is also recognized for its medicinal properties, acting as an antioxidant [33]. Additionally, it is effective in regulating the immune system and possesses anti-fungal, anti-cancer, and anti-lipid properties [34,35].

*Kelussia odoratissima*, commonly known as mountain celery, is an important edible and medicinal plant endemic to the central Zagros highlands and holds significant cultural importance for the Bakhtiari people (Fig 1c, d). This monotypic endemic plant can be found naturally in the northern part of Chaharmahal and Bakhtiari province, as well as in the northwest of Isfahan (Fereydunshahr), the eastern region of Khuzestan, and northern parts of Kohgiluyeh and Boyer-Ahmad provinces [36]. In traditional medicine, *K. odoratissima* is used to treat various disorders, including blood pressure, heart diseases, rheumatism, menstrual pain, and cholesterol management [37,38,39]. This medicinal plant contains essential oils with compositions such as (Z)-Ligustilide (76.45%), Unknown-A (4.47%), (E)-Ligustilide (2.57%), (Z)-Butylidene phthalide (2.37%), 5-pentyl cyclohexa-1,3-diene (1.57%), and kessane (0.77%) [40]. Additionally, the plant's alcoholic extract is rich in flavonoids (12.2 mg/g) and polyphenols (102 mg/g), contributing to its antioxidant properties [41]. *K. odoratissima* also exhibits various pharmacological benefits, including anti-allergic, analgesic, anti-diabetic, anti-inflammatory, and antimicrobial effects [42,43].

The province of Chaharmahal and Bakhtiari in Iran serves as a significant habitat and a primary distribution center for these species. A quantitative ethnobotanical study conducted by our team between 2020 and 2024 indicates that these two plants are among the most popular in the diet of the indigenous people in this region of Central Zagros (unpublished data). Additionally, these plants are widely recognized for their medicinal properties; the local community believes that mountain celery can cure up to 72 diseases, while shallots are

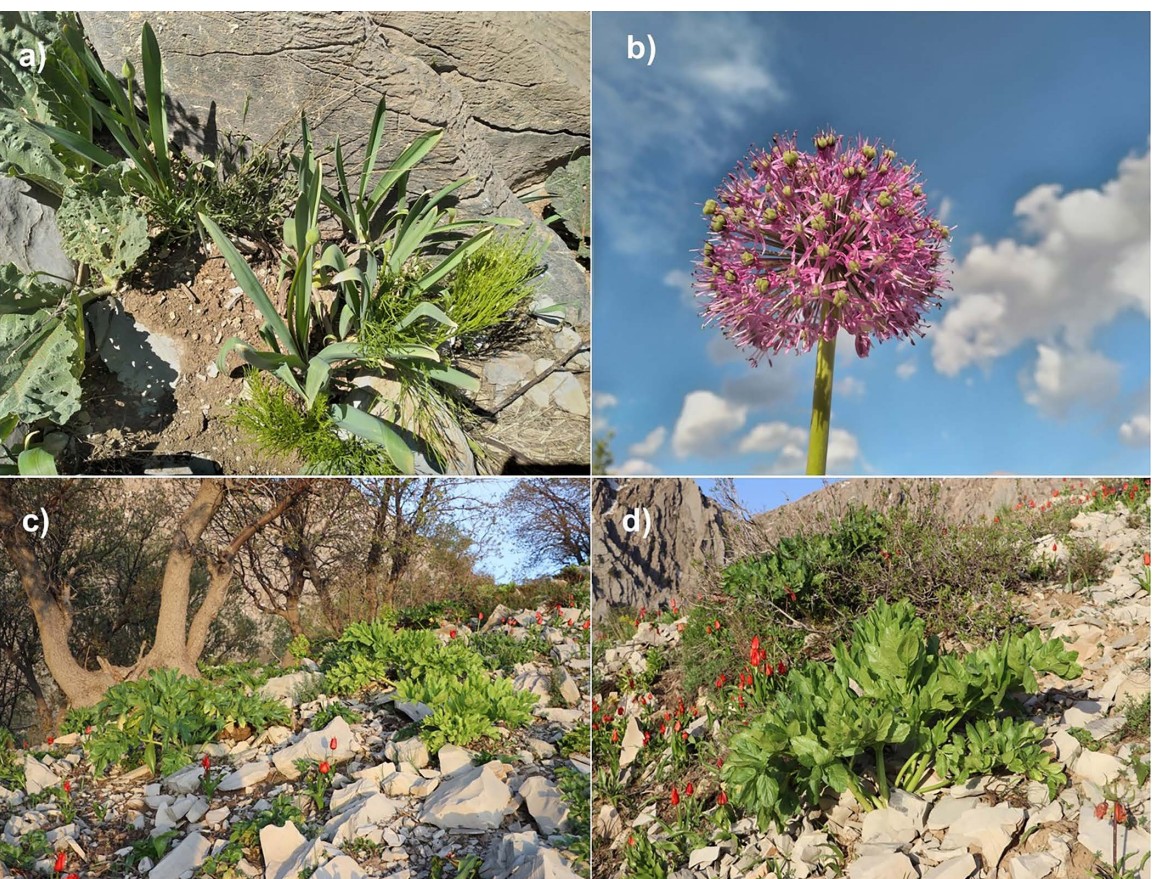

**Fig 1. Photographs of *Allium stipitatum* (a, b), and *Kelussia odoratissima* (c, d) in natural habitats.**

used to treat digestive and respiratory issues, diabetes, pain, and cardiovascular diseases. This popularity has unfortunately led to excessive harvesting in their natural habitats within the province. Moreover, our extensive research over recent years has shown that the indiscriminate harvesting and destruction of these species' habitats have resulted in the decline of many populations (Fig 2). Our decade-long research indicates that *A. stipitatum* is currently classified as "endangered (En)" in terms of conservation status, while *K. odoratissima* is classified as "critically endangered (CR)." There is a significant risk of these species disappearing in this province. For generations, these plants have been a vital source of income for the local population. However, the increasing human population and its environmental impact are now posing a serious threat to the survival of these species. Recent studies by Zeraatkar et al. [36] have demonstrated that *K. odoratissima* is on the verge of extinction. A recent study by Khajoei Nasab and Zeraatkar predicts that climate change, driven by global warming, is likely to significantly alter the distribution of certain medicinal and edible plant species in Chaharmahal and Bakhtiari province [44]. Given this information, researching the impact of climate change on the most popular plant species, which are at risk of extinction and hold significant traditional knowledge in the region, could be a valuable topic for researchers. Furthermore, these two species have adequate occurrence data available for conducting species distribution modeling studies.

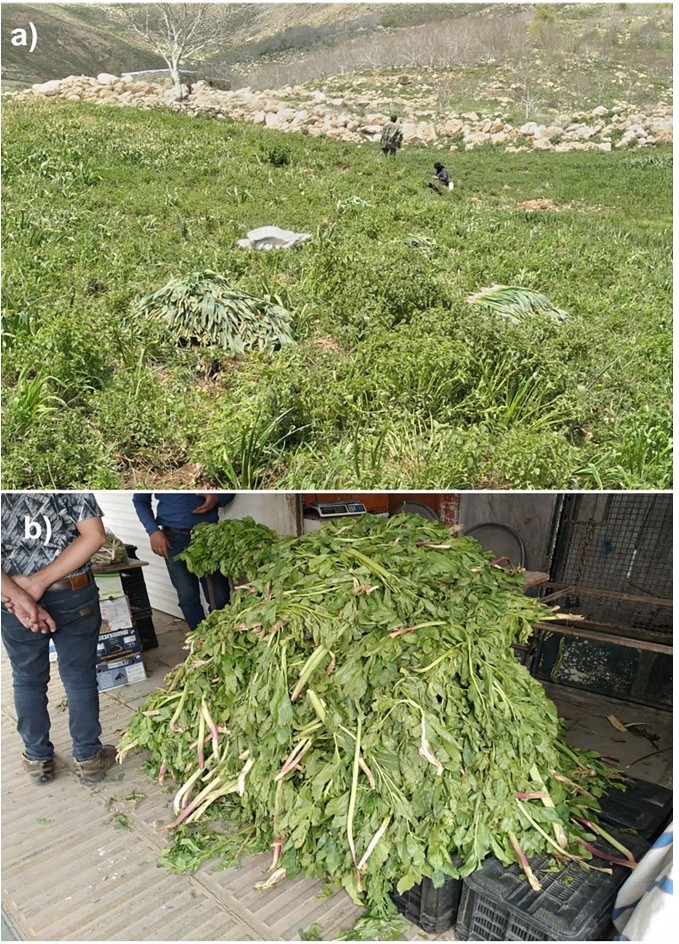

**Fig 2. Indiscriminate harvesting of wild populations of *Allium stipitatum* (a) and a wide supply of *Kelussia odoratissima* in the local markets of Shahrekord (b).**

Therefore, it is crucial to investigate the potential distribution of these species using species distribution models, considering these factors. The study aimed to achieve three main objectives: 1. Utilize the MaxEnt model to map the spatial distributions of *A. stipitatum* and *K. odoratissima* in Chaharmahal and Bakhtiari province based on current climate conditions. 2. Identify the key environmental factors that influence the distribution ranges of these plant species. 3. Predict changes in habitat distribution for these species under optimistic (RCP2.6) and pessimistic (RCP8.5) climate scenarios for the 2050s and 2070s.

## Materials and methods

### Study area

Chaharmahal and Bakhtiari Province is located in the western part of Iran and covers an area of approximately 16419 square kilometers. It is situated amidst the Zagros mountains, with geographical coordinates ranging from 31°9' to 32°48' N latitude to 49°30' to 51°26' E longitude. This province is recognized for its mountainous terrain, as it is part of Iran's central plateau. The climate in the region is diverse and can be categorized into five types: humid, very humid A, very humid B, Mediterranean, and semi-humid [45]. The annual rainfall is

about 560 mm, with Kuhrang receiving the highest at around 1800 mm. The average annual temperature ranges from 5 to 16°C, with an average of approximately 10°C.

## Occurrence data

The distribution points of the studied species in Chaharmahal and Bakhtiari province were collected through field sampling from their natural habitats. We also gathered occurrence records from specimens available in Herbarium D [acronyms according to 46], Flora of Chaharmahal and Bakhtiari province, and recent literature. To prevent spatial autocorrelation, we ensured that no spatial data was collected within one kilometer of existing occurrence points of the species. In total, we recorded 78 presence points for *A. stipitatum* and 15 presence points for *K. odoratissima* (Fig 3).

## Management of environmental variables

Current and future climate data, including 19 bioclimatic variables, were obtained from the WorldClim (https://www.worldclim.org/;Hijmans et al. 2005). The Digital Elevation Model (DEM) map was extracted from the raster layer available on www.worldgrids.org, and the

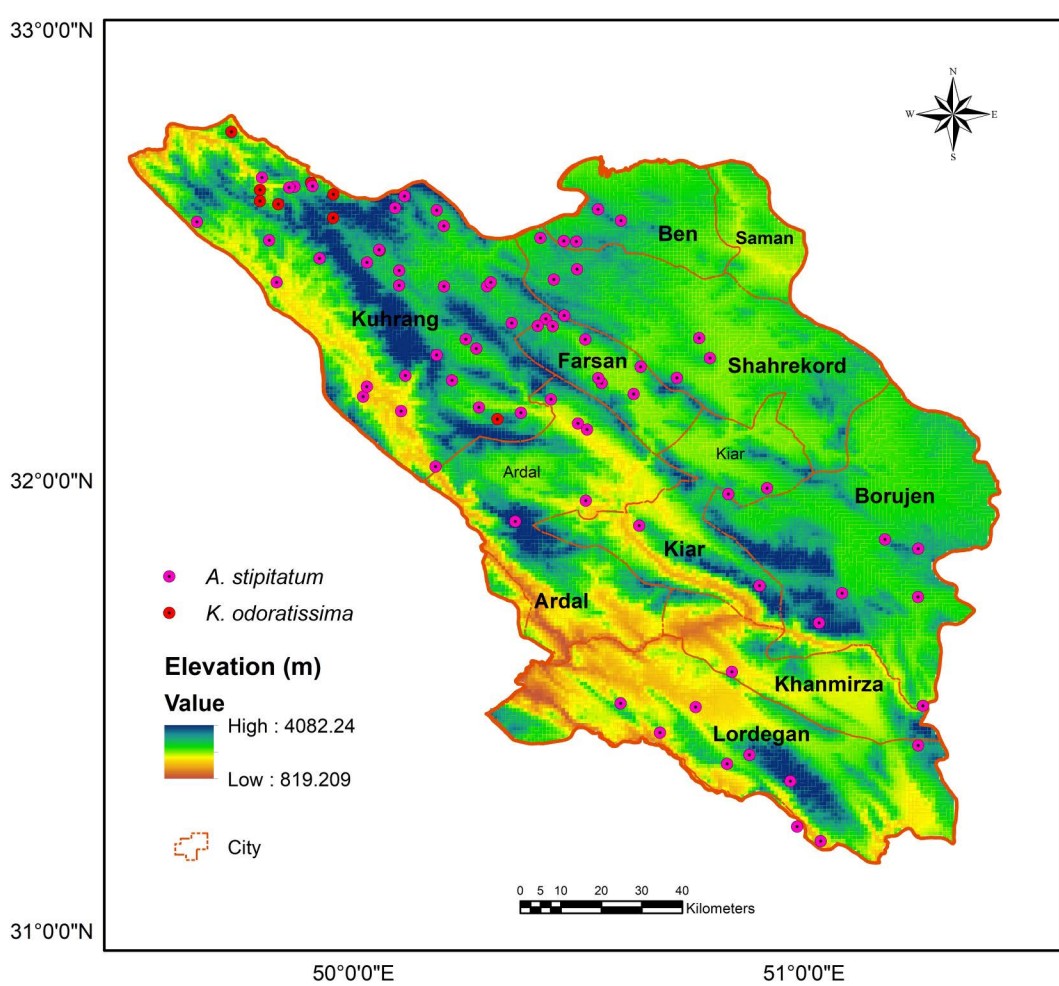

**Fig 3. Distribution map (presence data) of *Allium stipitatum* and of *Kelussia odoratissima* in Chaharmahal and Bakhtiari province (Using Arc-map 10.8.1 software (URL: https://www.arcgis.com/index.html).**

aspect and slope maps were generated using ArcGIS 10.8.1 software. To assess soil properties—such as texture, electrical conductivity (EC), pH, and organic carbon percentage—soil samples were collected from each location where the species were found, at depths of 0–30 cm below the surface. Maps for each property were created using the Inverse Distance Weighted interpolation (IDW) method within ArcGIS 10.8.1 software. To address collinearity among the variables, the variance inflation factor (VIF) was calculated. Highly correlated variables (VIF<10) were eliminated using the "USDM" package [47]. The results are available in the supplementary material (S1). As a result, 18 environmental variables were retained for *A. stipitatum* and 13 for *K. odoratissima* remain for model projection (Figs 4,5). The CCSM4 atmospheric general circulation model, along with optimistic (RCP2.6) and pessimistic (RCP8.5) climate scenarios, was used to evaluate the potential impact of climate change on species distribution in the 2050s and 2070s. The environmental variables in the raster layers were standardized with 30-second accuracy, which approximates one square kilometer, using ArcGIS 10.8.1 software.

## Modeling process and evaluation

The maximum entropy approach was utilized with MaxEnt v3.4.4k to estimate the current and potential future distribution of species [48]. We used 75% of the species occurrence records for

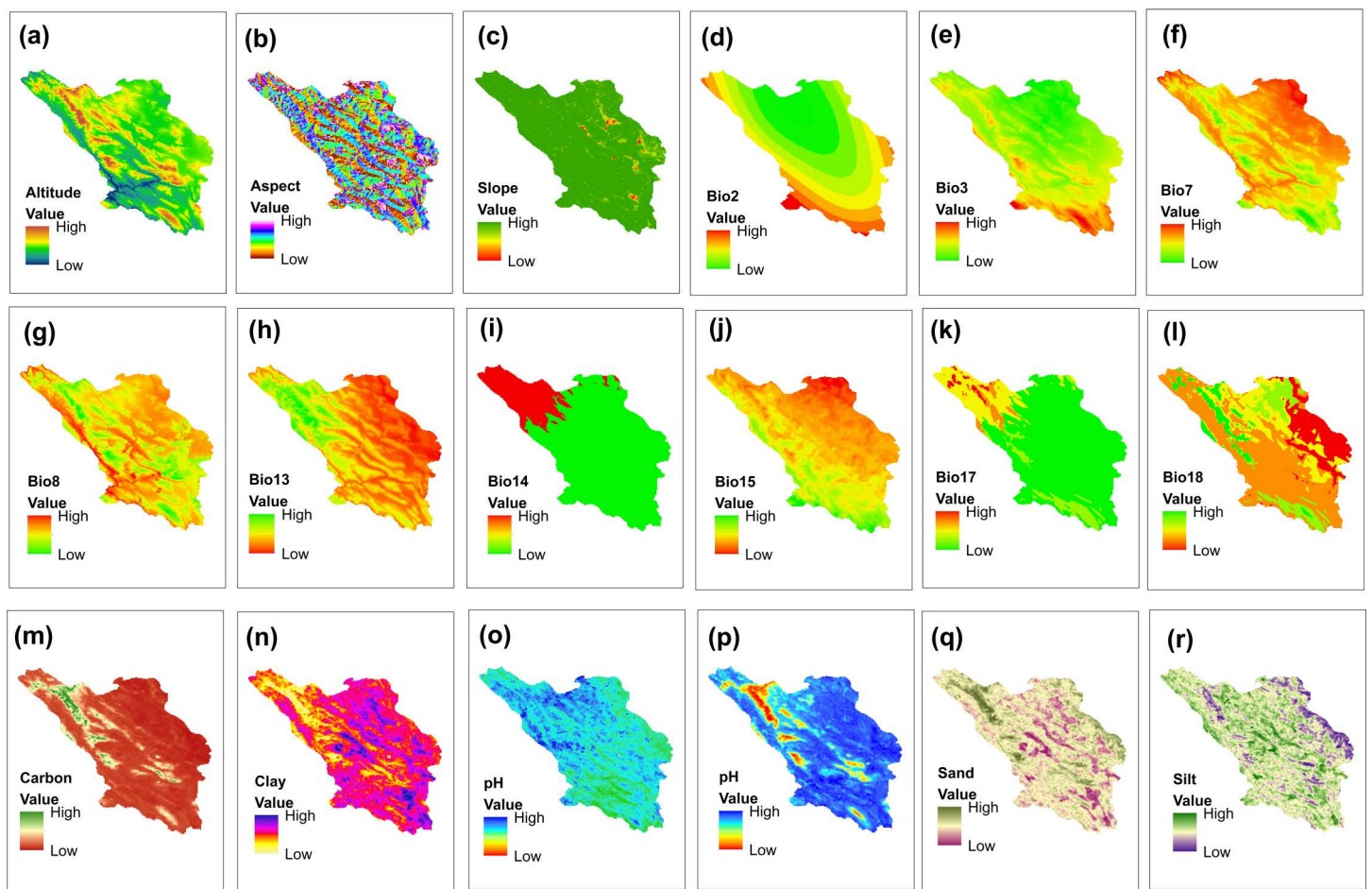

**Fig 4. Environmental variables related to the distribution of *Allium stipitatum* in Chaharmahal and Bakhtiari province.** (Using Arc-map 10.8.1 software (Using Arc-map 10.8.1 software (URL: https://www.arcgis.com/index.html).

model calibration and the remaining 25% for model testing. The model was run with 10 replicates, 10,000 background points, and a maximum of 5000 iterations. To evaluate the performance of the model, we employed the Area Under the Receiving Operator Curve (AUC) as a measure of accuracy that is not reliant on a specific threshold [49]. An AUC value of 0.5 indicates random prediction performance, while a value of 1 indicates high performance [50]. Furthermore, we used permutation importance to identify the most effective environmental variables.

## Results

### Model evaluation

Research findings indicate that the AUC is greater than 0.9 for *A. stipitatum* and *K. odoratissima*, which demonstrates the model's excellent performance in predicting the preferred habitats of the species under study.

### Key factors determining the potential species distribution

The distribution of *A. stipitatum* is primarily influenced by isothermality (Bio3) (32.4%), soil organic carbon (14.2%), and pH (11.8%) (Fig 6). In contrast, the potential distribution of the

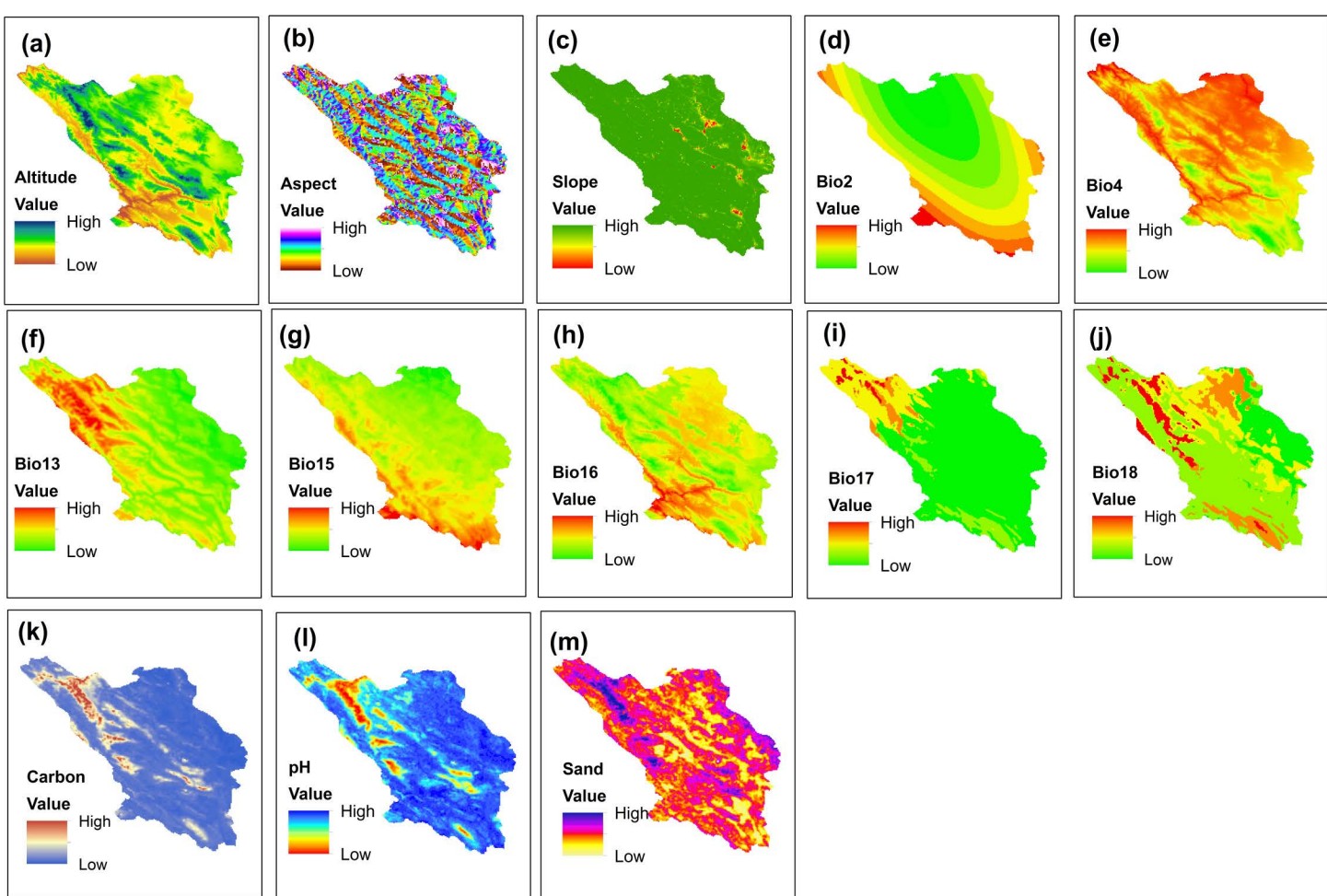

**Fig 5. Environmental variables related to the distribution of *Kelussia odoratissima* in Chaharmahal and Bakhtiari province (Using Arc-map 10.8.1 software (URL: https://www.arcgis.com/index.html).**

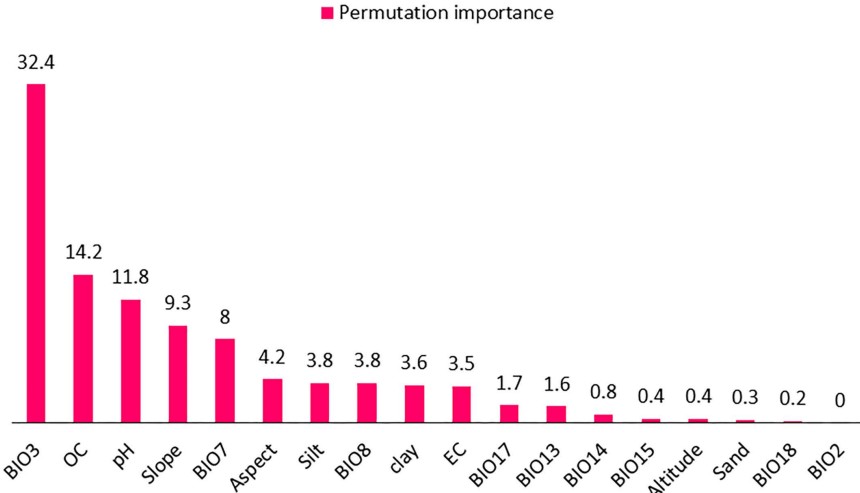

**Fig 6. The percentage of permutation importance for environmental factors used in SDM of *Allium stipitatum* in Chaharmahal and Bakhtiari province.**

*K. odoratissima* species is significantly impacted by soil organic carbon (53.5%), precipitation seasonality (Bio15) (12%), and precipitation of the wettest month (Bio13) (8.4%). These factors are illustrated in Fig 7.

## Current range of species

The potential habitat for the *A. stipitatum* species spans approximately 16,060 square kilometers, representing about 97.81% of the total area of the province. As shown in Fig 8, suitable habitats for the growth and distribution of this species are primarily located in Chelgerd, Farsan, Ardal, the west of Ben city, the north and west of Shahrekord, the southwest of Borujen,

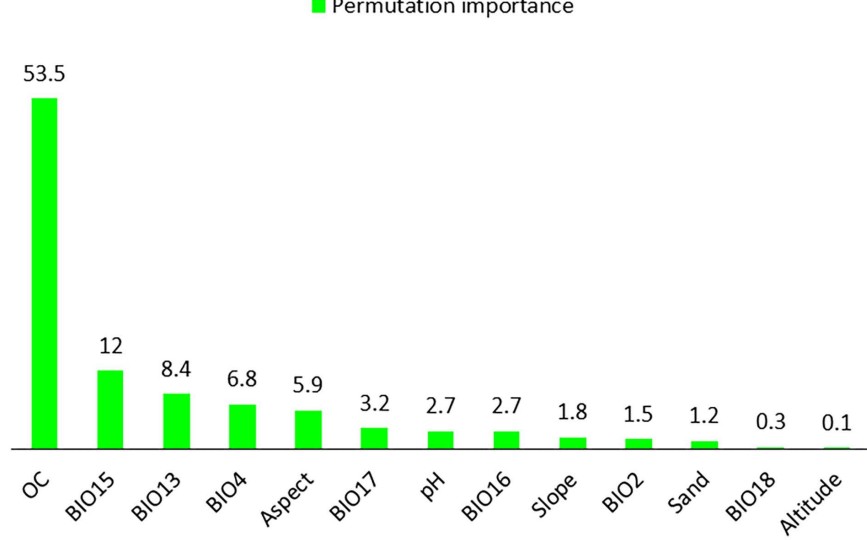

**Fig 7. The percentage of permutation importance for environmental factors used in SDM of *Kelussia odoratissima* in Chaharmahal and Bakhtiari province.**

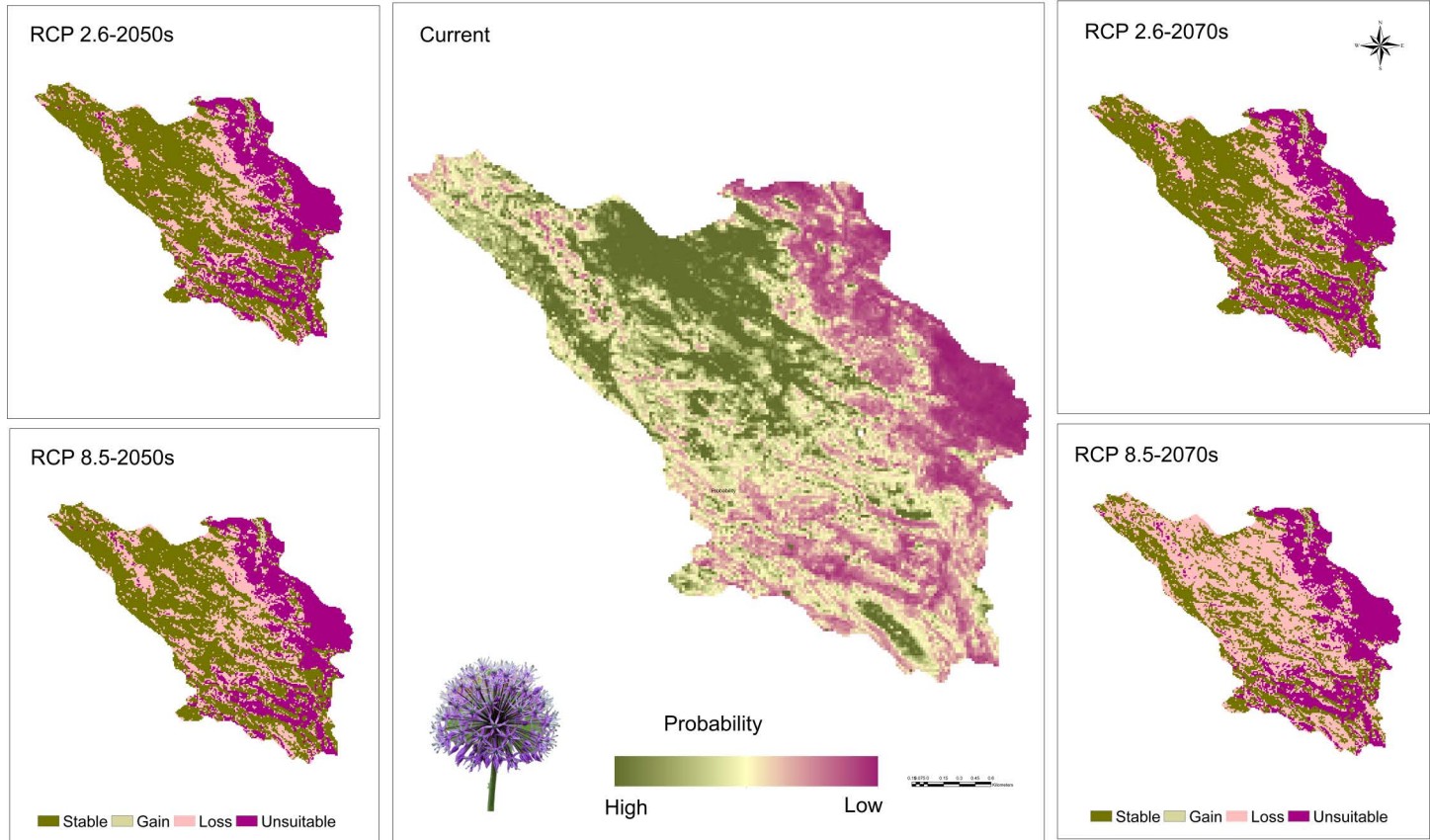

**Fig 8.  Map for potential current and future habitat suitability of *Allium stipitatum* in Chaharmahal and Bakhtiari province (Using Arc-map 10.8.1 software (URL:** https://www.arcgis.com/index.html**).**

the northern parts of Kiar and areas south of Lordegan city. In addition, around 4,413 square kilometers, or 26.87% of the province's total area, may support the *K. odoratissima* species (Fig 10). These areas are mainly located in Chelgerd, with additional suitable regions in Borujen, Ardal, Lordegan, and the limited regions of Farsan. The MaxEnt projections for *A. stipitatum* closely align with its current distribution, as illustrated in Fig 3 and 8. A comparison between the occurrence points we collected during field studies and the current potential distribution map of this species indicates that the areas identified by the model as potential habitats are fully consistent with our field observations (Figs 3 and 8). In contrast, for *K. odoratissima*, the potential habitats identified by the model are larger than the actual areas where this species occurs, as shown in Fig 10. Additionally, the model identified the regions of Ardal and Borujen as potential habitats for this species, even though it has not been reported in these areas to date (Figs 3 and 10).

## Future range of species

The distribution range of the *A. stipitatum* species is projected to decrease by approximately 29.60% to 55.12% due to climate change (Figs 8,9). Some suitable habitats will be lost in the central, southern, and northwestern parts of the study area. However, the Ben-Saman region in the northeastern part of the province, along with a small area in the southeastern part of Borujen, is expected for the growth and distribution of this species (Fig 8). As a result, it is likely that the species will migrate from the western to the eastern regions of the area. In terms of the

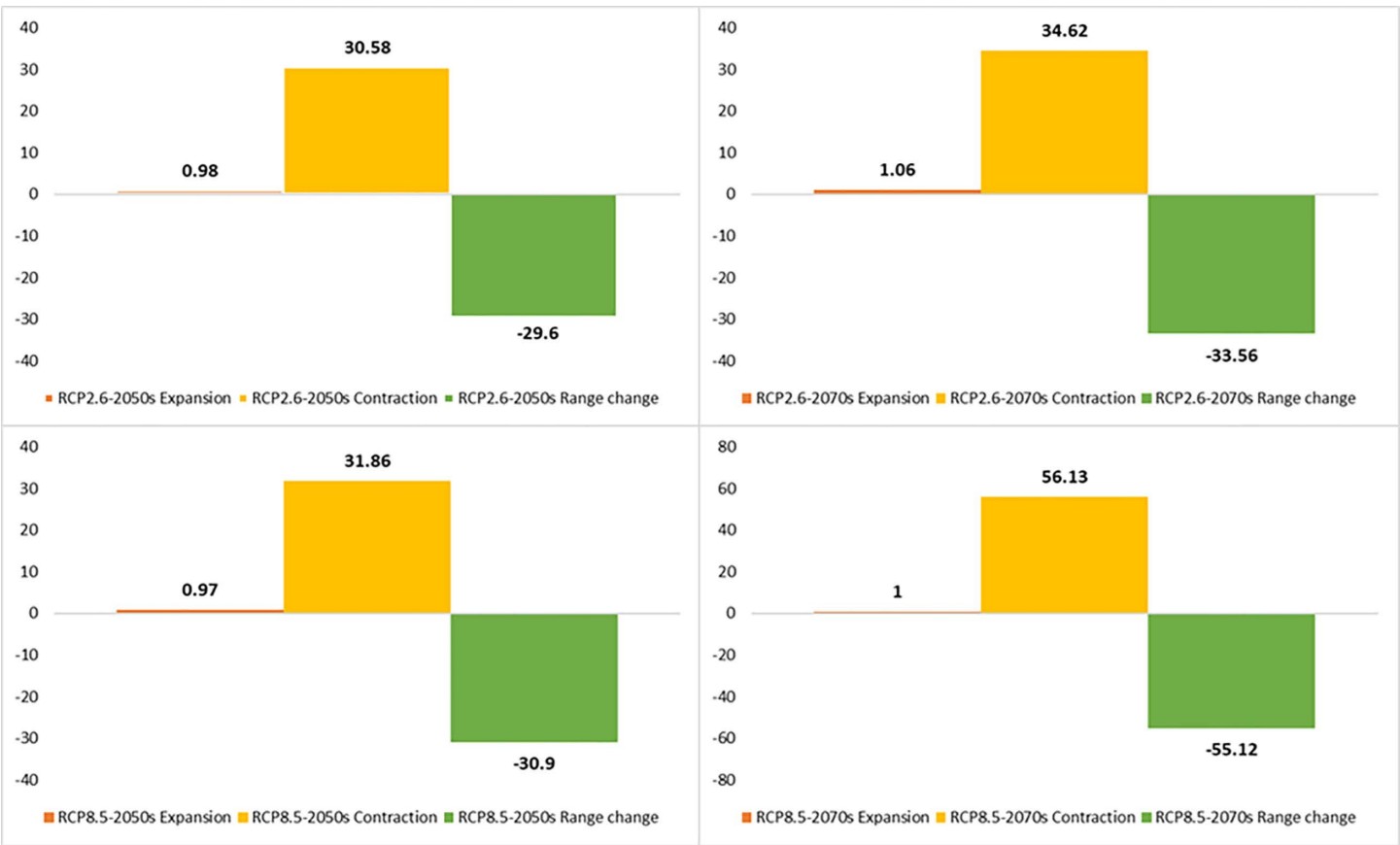

**Fig 9.  Percentage of the contraction, expansion, and range change of *Allium stipitatum* under optimistic (RCP 2.6) and pessimistic (RCP 8.5) greenhouse gas emission scenarios of the 2050s and 2070s.**

*K. odoratissima* species, the study indicates that it will be negatively affected by future climate change (Figs 10,11). It is estimated that between 26.72% and 71.61% of the species' preferred habitats will be lost. Notably, the 2050s are projected to experience a greater loss of favorable habitats for this species compared to the 2070s (Figs 10,11). The northwestern, western, and southern parts of the province are expected to be the hardest hit, resulting in a significant reduction in suitable habitats for this species. Conversely, less than 1% of new favorable habitat is expected to be gained. By the 2070s, however, it is anticipated that between 1.70% (RCP2.6) and 5.14% (RCP8.5) of new favorable habitats may be added, while existing favorable habitats for the species could decrease by between 31.86% (RCP8.5) and 53.43% (Figs 10,11).

## Discussion

The study used species distribution modeling to predict suitable habitats for *A. stipitatum* and *K. odoratissima*. The MaxEnt model demonstrated highly accurate predictive capability, with AUC values exceeding 0.9. Supporting this, other studies in the area have also confirmed the strong performance of the MaxEnt model in predicting plant species distribution [51,52]. Overall, the findings of this study indicate that both species are likely to be significantly affected by climate change.

The Persian shallot thrives in cold steppe areas and relatively high regions with adequate rainfall. This plant is resilient to severe winter cold and snowfall, making its bulbs resistant to

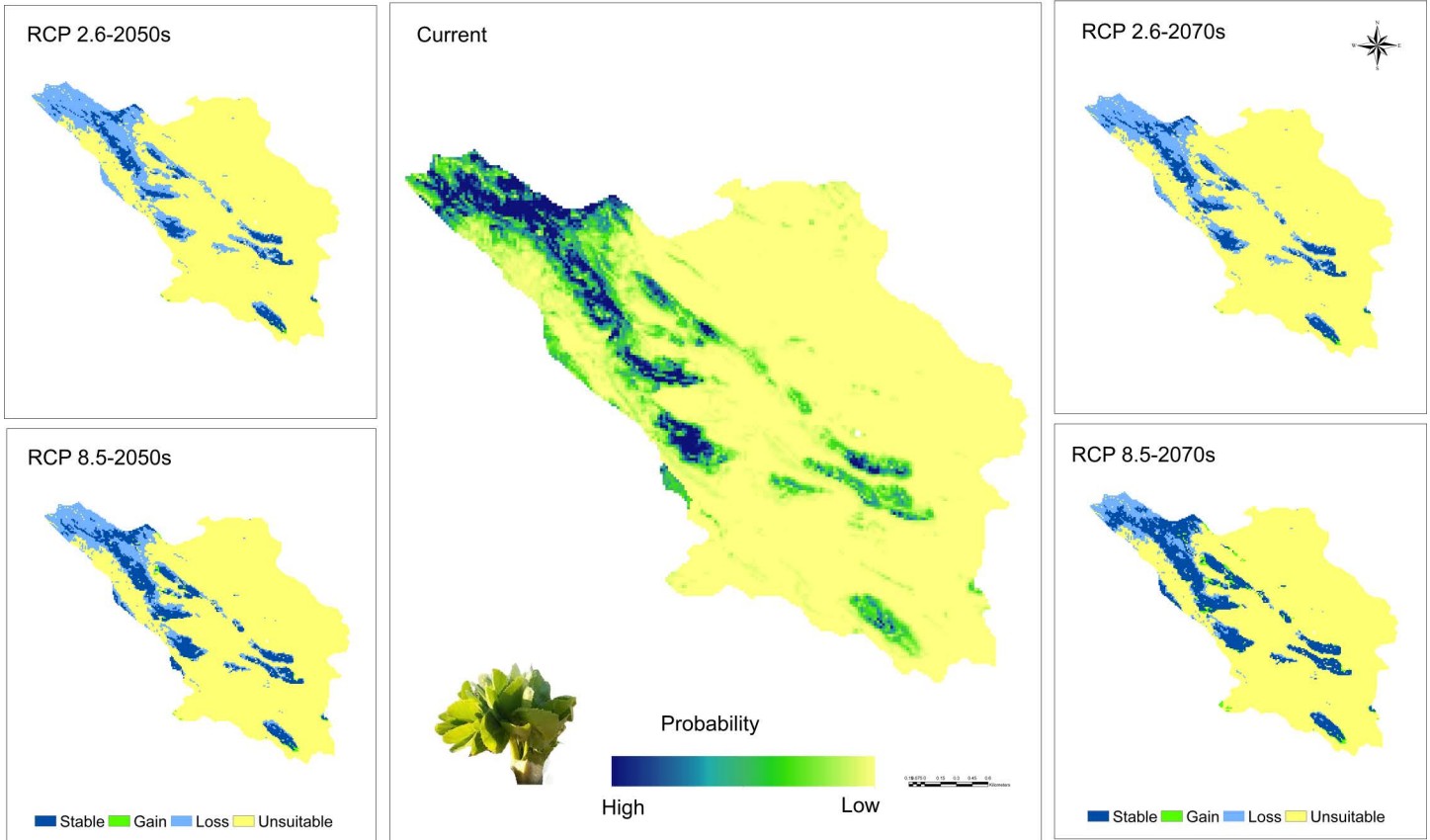

**Fig 10. Map for potential current and future habitat suitability of *Kelussia odoratissima* in Chaharmahal and Bakhtiari province (Using Arc-map 10.8.1 software (URL:** https://www.arcgis.com/index.html**).**

these conditions [53]. It typically grows on slopes and in the shade of trees and shrubs on the Iranian plateau [54]. The optimal average annual temperature for the Persian shallot ranges from 9 to 17 degrees Celsius. As temperatures rise, the growth period shortens, and the size of the bulbs decreases [55]. In its natural habitats, where winter rainfall occurs, the growth of the plant is influenced by environmental temperatures, which are not optimized outside this range. The study indicates that isothermality—measuring day and night temperature fluctuations compared to seasonal temperature variations—is a critical factor for the distribution of the plant. Specifically, the variations in day and night temperatures relative to annual summer and winter fluctuations are particularly important for the Iranian shallot. Temperature plays a crucial role as a limiting factor in the growth and distribution of plants. A study by Rahmanpour et al. [56] identified *A. stipitatum* as one of the least tolerant species among the five *Allium* species examined. This species was found to be withstand drought stress and dehydration. Additionally, Kafi et al.'s research [57] shows that when temperatures exceed optimal levels, some flowers may become sterile. Furthermore, the higher ambient temperatures cause the plant's thermal needs to be fulfilled in a shorter period. Consequently, the plant's flowering period is shortened, resulting in less fruit and seed production. Additionally, the duration of photosynthesis is also decreased, resulting in fewer photosynthetic materials being transported from the leaves to the seeds. This ultimately leads to a decrease in seed weight. As a consequence, the negative effects of climate change will reduce seed production in plants, resulting in fewer new seedlings in the future. The Persian shallot is expected to experience significant

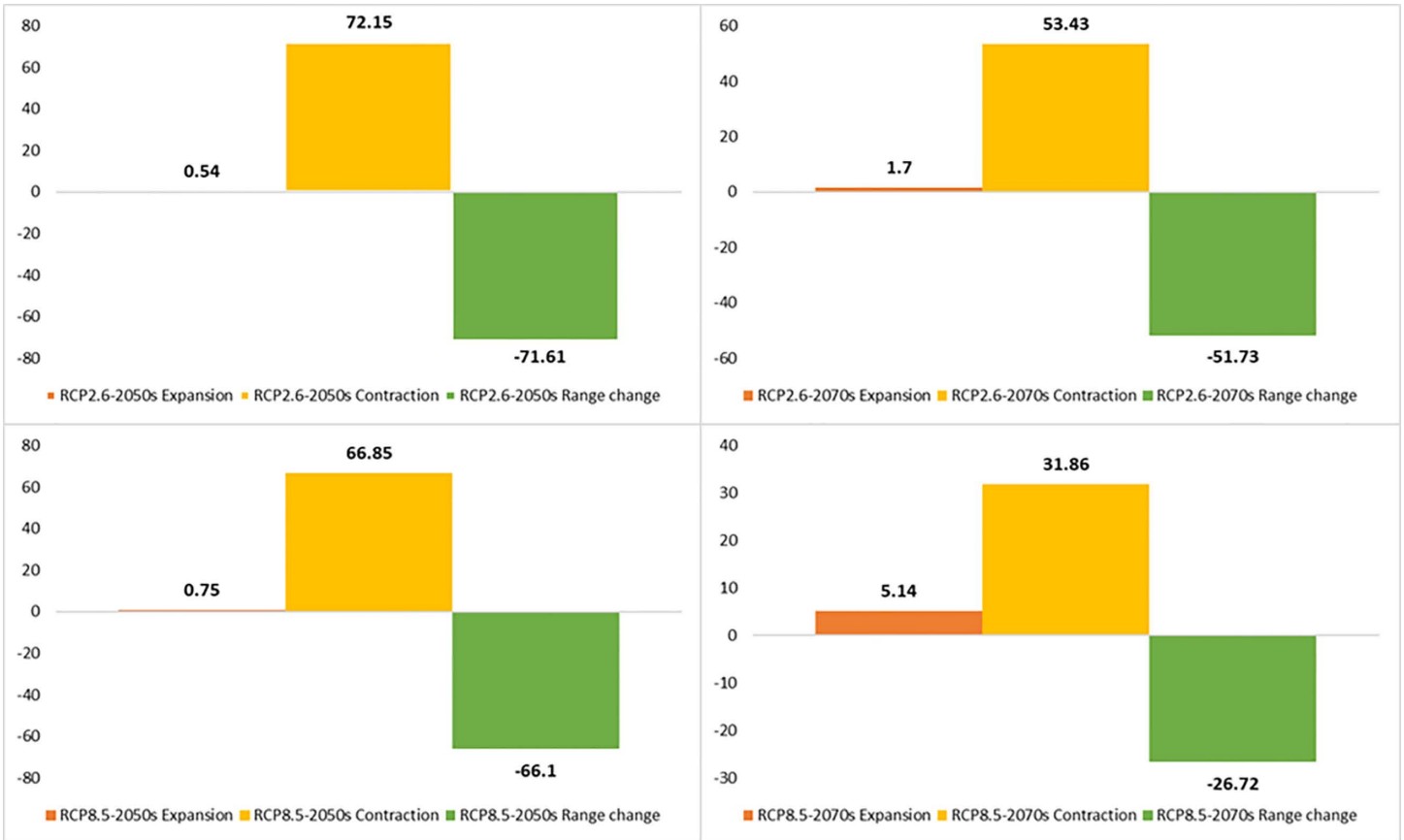

**Fig 11. Percentage of the contraction, expansion, and range change of *Kelussia odoratissima* under optimistic (RCP 2.6) and pessimistic (RCP 8.5) greenhouse gas emission scenarios of the 2050s and 2070s.**

losses due to the increasing temperatures associated with climate change. Similar findings were reported by Xie et al. [58], who discussed the role of Bio3 in the distribution of *Tapiscia sinensis* Oliver in China. Ali et al. [59] projected that the northward niche shift of *Monotheca buxifolia* (Falc.) A. DC. in the Hindu Kush-Himalayan mountainous (HHM) region would be primarily influenced by temperature and precipitation factors, including Bio3. Additionally, Jinga et al. [60] highlighted the significant impact of isothermality on the distribution of *Sclerocarya birrea* subspecies *caffra* in Africa.

In addition to temperature, two soil parameters—soil organic carbon and pH—have been identified as key factors influencing the distribution of this species. These soil factors significantly impact both the distribution and growth of the species. Similarly, Borhani and Sadeghzadeh [61] also highlighted the importance of soil factors, particularly pH, in relation to the species' distribution. Soil organic carbon is essential for soil quality, fertility, and agricultural profitability, as it enhances soil structure, water retention, and nutrient capacity [62]. Soil organic carbon impacts various chemical and physical processes in soil environments. It serves as a primary nutrient source for plants and provides a habitat for soil organisms [63]. Addis and Abebaw [64] highlighted the importance of soil organic carbon in the growth of *Allium sativum* L., indicating that it plays a crucial role in supplying nutrients, water, and suitable physical conditions. The species *A. stipitatum* species is typically found in soils with less than 5% soil organic carbon. Additionally, there is generally a negative correlation between soil salinity and the percentage of soil organic carbon [65]. Research on the ecology of this

species indicates that the plant can thrive and reproduce in low-salinity soils with an optimal level of organic carbon [61,66]. Climate change impacts soil organic carbon in two primary ways. First, rising temperatures accelerate the decomposition of organic matter in the soil, which reduces soil organic carbon levels and releases carbon dioxide, thereby contributing to global warming [67,68]. Second, shifts in precipitation patterns—such as increased droughts or heavy rainfall, can limit plant growth and reduce the input of organic matter into the soil, further decreasing soil organic carbon levels [69]. Consequently, species that are highly dependent on this environmental variable, like *A. stipitatum*, are likely to face significant disturbances due to climate change, resulting in a more restricted distribution. A study conducted by Hosseini et al. [30] has highlighted this trend regarding the habitat potential modeling of *Thymus transcaucasica* Ronniger. This research emphasizes the key role of soil organic carbon in the distribution of this species. Similarly, Jobbágy and Jackson [70] found that soil carbon is an important factor for plants in forest ecosystems. Additionally, previous studies have identified soil carbon as one of the most crucial factors in determining the distribution of nectar-producing species of *Nepeta* in Iran [71].

Soil pH is a crucial factor in determining various chemical and biochemical processes within the soil [72]. *A. stipitatum* thrives in habitats with alkaline soils, typically having a pH range of 8 to 8.2. A study by Allahmoradi et al. [73] investigated the habitat characteristics of Iranian shallots and underscored the significant impact of soil pH on the distribution of this species. The study also confirmed the presence of *A. stipitatum* in alkaline soils. Furthermore, climate change and fluctuations in soil parameters can lead to considerable alterations in the types of vegetation found in different regions [74]. Piri Sahragard and Zare chahouki [75] showed that pH had the greatest impact on the distribution of *Artemisia sieberi* Besser. As was found in another study by Amindin et al. [76], also reported a direct relationship between the predictions of the distribution of *Fritillaria imperialis* L., with pH amount. Similarity, Zare et al. [77] showed a positive relationship between *Dorema ammoniacum* D Don. distribution in the rangelands of central Iran and soil pH.

It is predicted that heavy rains, which result from global warming, may cause to soil erosion and acidification in high mountain areas [78]. Conversely, downstream regions may become more alkaline due to increased drought conditions brought on by climate change [78]. A study conducted by Sun et al. [79], found that climate change has led to soil acidification in alpine pastures, followed by alkalinization in the alpine steppes of the Tibetan Plateau. Our study reached a similar conclusion, indicating that the areas most likely to lose their suitable habitats due to climate change are primarily the high-altitude pastures. As the soil becomes more acidic, these regions will become less suitable for the growth of Iranian shallots. However, the Ben and Saman region, as well as parts of Borujen, are high steppes that currently exhibit the highest levels of soil acidity, as illustrated in Fig 1. Therefore, it is reasonable to expect that these areas may become suitable locations for the growth and distribution of this species in the future.

The research findings indicate that *K. odoratissima* is more susceptible to damage from future climate changes than *A. stipitatum*. This vulnerability can be attributed to three main factors. Firstly, *K. odoratissima* has a more limited distribution within the study area, primarily inhabiting high mountain regions in the northern part of the province, particularly around Kuhrang. In contrast, *A. stipitatum* is found in a variety of habitats across the region, which suggests that it is more adaptable to different environments and less prone to environmental stresses compared to *K. odoratissima*. Secondly, the research model identified two key climatic factors, Bio15 and Bio13, as the primary constraints on the distribution of the species. Similar findings were observed in a study on *Acmella radicans* (Jacquin) R.K. Jansen conducted in China, indicating that the distribution of this species depends on Bio13 [80]. Furthermore,

this factor has a significant impact on the suitable area for *Larix gmelinii* (Rupr.) Rupr. [81]. Another precipitation-dependent factor, Bio15, plays a crucial role in the distributions of *Saposhnikovia divaricata* (Turcz.) Schischk. [82] and *Alpinia officinarum* Hance [83]. *K. odoratissima* typically thrives in snow catchment areas at altitude ranging from 1,920–3,100 meters, with an average annual rainfall of 450 mm. Future climate changes are expected to lead to increased rainfall in higher elevations, exceeding the optimal limits for *K. odoratissima*. This excessive rainfall may impair the plant's ability to adapt environmental changes and disrupt its interactions with pollinators, which could hinder its reproductive success. As global temperatures rise, the anticipated increases in rainfall may also affect plant-pollinator interactions, with potential consequences for both ecological and economic systems [84]. Excessive rainfall can disrupt pollen transfer and impede the reproductive processes of flowering plants through various mechanisms [85]. Similar to Iranian shallots, *K. odoratissima* depends on the availability of organic carbon in the soil. A shortage of this essential organic carbon could threaten the survival of this species. In the future, rising temperatures may cause populations at lower elevations to decline, and this could present challenges for cold-adapted mountain species like *K. odoratissima* [22]. Additionally, increased competition from species migrating to higher elevations due to climate change may disrupt the functioning of ecosystems [86]. Mountain species could also experience the "summit trap phenomenon," which hinders their ability to migrate and places them at greater risk [87].

Studies on *Hordeum bulbosum* L., *Stipa hohenackeriana* Trin & Rupr, and *Carataegus azarolus* L. in Chaharmahal and Bakhtiari province indicate that these species are likely to experience a reduction in their distribution due to climate change, possibly losing their suitable habitats [52,88,89]. Modeling studies conducted on various plant species across different regions of Iran suggest that global warming could have detrimental effects on their distribution. For example, the modeling studies conducted on various plant species in different regions of Iran indicate that global warming may have destructive effects on their distribution. For instance, Mirhashemi et al. [90] noted that Brant's oak (*Quercus brantii* Lindl.) is expected to largely disappear in Ilam province as a result of future climate changes. Safaei et al. [91] reported that nearly half of the populations of this species could be lost in the Zagros forests of Iran. Additionally, Khajoei Nesab et al. [92] predicted a decline and possible extinction of certain endemic species of the *Allium* genus in the northern, northwestern, and northeastern regions of Iran due to climate change. Similar studies from other parts of the world indicate a potential reduction in the distribution of many plant species [93,94,95,96]. Consequently, future climate changes present a significant threat to the survival of various plant species.

Our field investigations reveal that the distribution area of this plant in Chaharmahal and Bakhtiari province was once much larger than it is today. However, over time, their distribution has significantly decreased due to excessive harvesting and grazing. The natural habitats of *K. odoratissima* have been largely destroyed, and this species can now only be found in inaccessible areas [36]. Therefore, the results of this research highlight the urgent need for immediate solutions to protect these species.

## Mitigation strategies

It is essential to implement conservation efforts, both in situ (in their natural habitats) and ex situ (outside their natural habitats), to prevent the decline of these species. Currently, less than 2% of their natural habitats are located within the four regions managed by Iran's Environmental Protection Organization, making immediate protective measures crucial. These measures may include creating enclosures and establishing reserves for medicinal plants and protected areas. To tackle population declines caused by overharvesting, the Environmental Protection Organization should enforce a ban on animal grazing and the collection of these

species for medicinal purposes. Additionally, cultivating these plants in fields can support the sustainable use of their medicinal and edible properties. Propagation outside of natural habitats, such as in botanical gardens or research centers, can also be effective. Once these plants are cultivated, they can be reintroduced to strengthen wild populations. Furthermore, maintaining these species in gene banks is vital for conservation under extreme conditions. The preserved seeds or plant parts can assist in future habitat restoration or be cultivated in botanical gardens if natural conditions become unsuitable.

## Conclusions

In this study, we examined how climate change is impacting the distribution and habitat suitability of two valuable edible-medicinal species in the Chaharmahal and Bakhtiari provinces. Although these species are cultivated in certain areas of the province, they are also being harvested from their natural habitats by the local community and sold at very high prices in local markets. This poses a significant issue, as mountain celery represents a unique genus found only in Iran and has a limited distribution. Human activities, such as uncontrolled grazing and overharvesting, threaten the survival of this important species. Additionally, while this area serves as a crucial distribution center for Iranian shallots, excessive harvesting may lead to a depletion of wild populations in the coming years. Our findings indicate that climate change is likely to significantly impact the distribution and potential habitat suitability of these species, leading to important ecological and socio-economic consequences. Therefore, our study emphasizes the urgent need for conservation efforts to prevent their extinction and preserve their habitats. Furthermore, this research pinpoints potential suitable habitats for planting these species. It's worth noting that less than two percent of the suitable habitats for these species are located in protected environmental areas. To prevent extinction, conservation officials are encouraged to use this research to develop effective preservation strategies.

## Supporting information

**Table S1.** The variance inflation factors (VIFs) of the remained variables of *Allium stipitatum* and *Kelussia odoratissima.*
(DOCX)

## Acknowledgments

The authors wish to thank Iran National Science Foundation (INSF) for supporting the authors in conducting the current research study.

## Author contributions

**Conceptualization:** Farzaneh Khajoei Nasab, Amin Zeraatkar.

**Data curation:** Farzaneh Khajoei Nasab, Amin Zeraatkar.

**Formal analysis:** Farzaneh Khajoei Nasab.

**Investigation:** Farzaneh Khajoei Nasab, Amin Zeraatkar.

**Methodology:** Farzaneh Khajoei Nasab.

**Project administration:** Amin Zeraatkar.

**Software:** Farzaneh Khajoei Nasab.

**Supervision:** Amin Zeraatkar.

**Validation:** Farzaneh Khajoei Nasab.

**Visualization:** Farzaneh Khajoei Nasab.

**Writing – original draft:** Farzaneh Khajoei Nasab, Amin Zeraatkar.

**Writing – review & editing:** Farzaneh Khajoei Nasab, Amin Zeraatkar.

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
