## [Decision Letter · Decision Letter 0]

8 Jan 2025

PONE-D-24-55210Two plant species, Allium stipitatum and Kelussia odoratissima, will suffer due to climate change in Central ZagrosPLOS ONE

Dear Dr. Khajoei Nasab,

Thank you for submitting your manuscript to PLOS ONE. After careful consideration, we feel that it has merit but does not fully meet PLOS ONE’s publication criteria as it currently stands. Therefore, we invite you to submit a revised version of the manuscript that addresses the points raised during the review process.

(Reviewer 1)

The english writing style is not acceptable

you must to compare the model output with field evidence

How did you validate the model output?

The discussion section must be improved

the title must be rewrite

(Reviewer 2)

The manuscript is well-written and effectively elucidates the impact of climate change on two significant plant species in Iran. I have some minor queries:

1. Selection of Plant Species: Why were only these particular plants chosen for the study? Under similar conditions, are other plants more resistant to climate change?

2. Vulnerability Comparison: Why are these plants more susceptible to climate change compared to other species in the area? Do the climate factors discussed in this study have a lesser impact on other regional plants?

3. Historical Climate Data: Is there data indicating changes in climate factors such as temperature, rainfall, and soil conditions over past decades?

4. Future Climate Projections: What are the anticipated changes in climate conditions in the future compared to the present scenario?

5. Mitigation Strategies: Considering that climate change is a global phenomenon, what potential solutions could be implemented in these areas to mitigate its effects?

We look forward to receiving your revised manuscript.

Kind regards,

Sara Hemati

Academic Editor

PLOS ONE

“This research was done as part of the Post Doctorate Program of F. KHN. under the supervision of A.Z which was financially supported by Iran National Science Foundation: INSF (Grant No. 4020273).”

7. We note that Figures 3,4,5,8 and 10 in your submission contain [map/satellite] images which may be copyrighted. All PLOS content is published under the Creative Commons Attribution License (CC BY 4.0), which means that the manuscript, images, and Supporting Information files will be freely available online, and any third party is permitted to access, download, copy, distribute, and use these materials in any way, even commercially, with proper attribution. For these reasons, we cannot publish previously copyrighted maps or satellite images created using proprietary data, such as Google software (Google Maps, Street View, and Earth). For more information, see our copyright guidelines: http://journals.plos.org/plosone/s/licenses-and-copyright.

a. You may seek permission from the original copyright holder of Figures 3,4,5,8 and 10 to publish the content specifically under the CC BY 4.0 license. 

Reviewers' comments:

Reviewer's Responses to Questions

**Comments to the Author**

1. Is the manuscript technically sound, and do the data support the conclusions?

Reviewer #1: No

Reviewer #2: Yes

2. Has the statistical analysis been performed appropriately and rigorously? 

Reviewer #1: Yes

Reviewer #2: N/A

3. Have the authors made all data underlying the findings in their manuscript fully available?

Reviewer #1: No

Reviewer #2: Yes

4. Is the manuscript presented in an intelligible fashion and written in standard English?

Reviewer #1: No

Reviewer #2: Yes

5. Review Comments to the Author

Reviewer #1: The english writing style is not acceptable

you must to compare the model output with field evidence

How did you validate the model output?

The discussion section must be improved

the title must be rewrite

Reviewer #2: The manuscript is well-written and effectively elucidates the impact of climate change on two significant plant species in Iran. I have some minor queries:

1. Selection of Plant Species: Why were only these particular plants chosen for the study? Under similar conditions, are other plants more resistant to climate change?

2. Vulnerability Comparison: Why are these plants more susceptible to climate change compared to other species in the area? Do the climate factors discussed in this study have a lesser impact on other regional plants?

3. Historical Climate Data: Is there data indicating changes in climate factors such as temperature, rainfall, and soil conditions over past decades?

4. Future Climate Projections: What are the anticipated changes in climate conditions in the future compared to the present scenario?

5. Mitigation Strategies: Considering that climate change is a global phenomenon, what potential solutions could be implemented in these areas to mitigate its effects?

6. PLOS authors have the option to publish the peer review history of their article (what does this mean? ). If published, this will include your full peer review and any attached files.

**Do you want your identity to be public for this peer review?** For information about this choice, including consent withdrawal, please see our Privacy Policy .

Reviewer #1: No

Reviewer #2: **Yes: ** Ashraf Ali

---

## [Author Response · Author response to Decision Letter 1]

21 Feb 2025

Dear editor-in-chief

Thank you very much for giving us the opportunity to submit a revised draft of the manuscript “Assessing the impact of global warming on the distributions of Allium stipitatum and Kelussia odoratissima in the Central Zagros using a MaxEnt model” for publication in the PLOS One. We have incorporated most of the suggestions made by you on the manuscript. Images 3, 4, 5, 8, and 10 are not satellite images, etc. We generated them using ArcMap 10.8.1 software. Additionally, we edited the captions for these images, which are highlighted in purple. Please see the attachment.

Sincerely yours

Authors

Reviewers' Comments to the Authors:

COMMENTS TO THE AUTHOR:

Reviewer#1

Authors response:

We would like to express our sincere gratitude for taking the time to review our manuscript for publication in PLOS One. Your feedback, along with that of the other reviewers, has been invaluable in helping us enhance the original framework. We have made significant modifications based on your suggestions, which are highlighted in yellow throughout the text. Thank you once again for your valuable contributions.

The english writing style is not acceptable

Author's response:

We have edited all the text in terms of English writing.

you must to compare the model output with field evidence

Author's response:

Thank you for your valuable comment. We have added this content to the results section.

The MaxEnt projections for A. stipitatum closely align with its current distribution, as illustrated in Figures 3 and 8. A comparison between the occurrence points we collected during field studies and the current potential distribution map of this species indicates that the areas identified by the model as potential habitats are fully consistent with our field observations (Figs. 3 and 8). In contrast, for K. odoratissima, the potential habitats identified by the model are larger than the actual areas where this species occurs, as shown in Figure 10. Additionally, the model identified the regions of Ardal and Borujen as potential habitats for this species, even though it has not been reported in these areas to date (Figs. 3 and 10).

How did you validate the model output?

Author's response:

As previously mentioned in the "Modeling Process and Evaluation" section of the manuscript, the validation of the model output was conducted using the Area Under the Receiver Operating Characteristic Curve (AUC). Most of the researchers are using AUC to validate the output of Maxent. AUC values are listed in the manuscript for each species.

Please see these sentences:

To evaluate the performance of the model, we employed the Area Under the Receiving Operator Curve (AUC) as a measure of accuracy that is not reliant on a specific threshold [47]. An AUC value of 0.5 indicates random prediction performance, while a value of 1 indicates high performance [48].

The discussion section must be improved

Author's response:

Thank you. We have improved this section.

the title must be rewrite

Author's response:

Thank you. We rewrote this section.

Reviewer#2

The manuscript is well-written and effectively elucidates the impact of climate change on two significant plant species in Iran. I have some minor queries:

Authors response:

We would like to express our sincere gratitude for taking the time to review our manuscript for publication in PLOS One. Your feedback, along with that of the other reviewers, has been invaluable in helping us enhance the original framework. We have made significant modifications based on your suggestions, which are highlighted in green throughout the text. Thank you once again for your valuable contributions.

1. Selection of Plant Species: Why were only these particular plants chosen for the study? Under similar conditions, are other plants more resistant to climate change?

Author's response:

Thank you for your valuable question. We considered several criteria when selecting the species for our study:

1. Ethnobotanical studies can greatly assist conservation efforts by identifying the most popular plant species in a given region. Conducting comprehensive quantitative ethnobotanical studies by our team in the province from 2020 to 2024, which included various indigenous groups, revealed that these two plant species are the most favored in the local diet. Additionally, they possess numerous medicinal properties within traditional knowledge, making them compelling candidates for conservation studies.

2. Our focus was not limited to ethnobotanical studies alone; extensive floristic research we conducted in recent years indicated that these two species are highly endangered due to their popularity. Thus, their status as endangered species was another key factor in our selection process.

3. We also aimed to study species with significant economic value in the region. Our research demonstrated that these two species command high prices in local markets. For instance, in the spring of this year, the mountain celery species was sold at an astonishing 5 million tomans per kilogram, making it the most expensive plant in local markets. This high economic value incentivizes harvesters, which contributes to the degradation of these species’ natural habitats.

4. We sought species primarily distributed in Iran, particularly within this province, to evaluate the impact of climate change on populations residing at the center of distribution for these species in Iran. By examining various resources, including the floras of Iran and the province, we determined that these two species are optimal choices for our study.

5. There are several standards for conducting modeling studies that impose limitations. For example, many endemic species in this province are at significant risk of extinction, but due to insufficient occurrence data, we are unable to model their distribution. Numerous species have only been documented at a single location, which also restricts modeling opportunities. Hence, we decided to focus our study on these two species. It's important to clarify that our intention is not to imply that other species are more resilient to climate change. Each species has unique ecological requirements and responds differently to climate change.

To eliminate any ambiguity, we have included our reasoning in the introduction, which is highlighted in green. If you have any further questions or need clarification, please do not hesitate to reach out, and we will do our best to assist you.

2. Vulnerability Comparison: Why are these plants more susceptible to climate change compared to other species in the area? Do the climate factors discussed in this study have a lesser impact on other regional plants?

Author's response:

As stated earlier, our aim in conducting this study is not to suggest that other plant species in the region are less sensitive to climate change or not vulnerable at all. In the Introduction and Discussion sections, we noted that modeling studies predict future climate change will negatively affect the distribution of other species in this province. For example:

Studies conducted on Hordeum bulbosum L., Stipa hohenackeriana Trin & Rupr, and Carataegus azarolus L. in the Chaharmahal and Bakhtiari province also suggest that these species will undergo a reduced distribution in response to climate change, potentially losing their suitable habitats [74, 50, 75]. A recent study by Khajoei Nasab and Zeraatkar predicts that climate change, driven by global warming, is likely to significantly alter the distribution of certain medicinal and edible plant species in Chaharmahal and Bakhtiari province [?].

This province is home to a wide variety of plant species, and assessing the vulnerability of all these species is a lengthy and costly endeavor that is challenging to fund. As a result, we focused on evaluating the most significant species by establishing specific criteria. We hope that in the near future, with adequate funding and time, we will be able to study the impact of climate change on the distribution of all plant species in this province.

3. Historical Climate Data: Is there data indicating changes in climate factors such as temperature, rainfall, and soil conditions over past decades?

Author's response:

Unfortunately, there is insufficient historical information regarding climatic factors in the study area due to the limited data from the country's synoptic and meteorological stations. Over the past ten years, only four weather stations in the province provided data, which does not adequately represent the entire region. However, global databases such as WorldClim offer some historical climatic information available at the following URLs:

https://www.worldclim.org/data/worldclim21.html;
https://www.worldclim.org/data/monthlywth.html.

Similarly, soil information is also quite limited in domestic research centers. Global databases like the Harmonized World Soil Database (v1.2) (https://www.fao.org/soils-portal/data-hub/soil-maps-and-databases/harmonized-world-soil-database-v12/en/) and the Soil Geographic Database — World (https://www.isric.org/soil-geographic-databases-world) contain a combination of data on current soil conditions as well as historical information covering periods from 1961-1985 or 1971-1981.

4. Future Climate Projections: What are the anticipated changes in climate conditions in the future compared to the present scenario?

Author's response:

The projected distribution range of A. stipitatum is expected to decrease by 29.60% to 55.12% due to future climate change compared to current conditions. This decline will particularly affect the central, southern, and northwestern regions. However, there may be potential for this species to migrate from west to east, as the Ben-Saman region in the northeast and a small portion of southeastern Borujen could become suitable habitats.

In contrast, K. odoratissima is anticipated to experience significant habitat loss, estimated between 26.72% and 71.61%. The greatest losses are expected in the 2050s rather than the 2070s, with the northwestern, western, and southern parts of the province facing the most severe impacts. Although there may be a slight gain of less than 1% in new favorable habitats, by the 2070s, only between 1.70% (under RCP2.6) and 5.14% (under RCP8.5) of new habitats are expected, alongside a decline of 31.86% to 53.43% in existing habitats.

5. Mitigation Strategies: Considering that climate change is a global phenomenon, what potential solutions could be implemented in these areas to mitigate its effects?

Author's response:

Thank you for your valuable comment. It is essential to implement conservation efforts, both in situ (in their natural habitats) and ex situ (outside their natural habitats), to prevent the decline of these species. Currently, less than 2% of their natural habitats are located within the four regions managed by Iran's Environmental Protection Organization, making immediate protective measures crucial. These measures may include creating enclosures and establishing reserves for medicinal plants and protected areas. To tackle population declines caused by overharvesting, the Environmental Protection Organization should enforce a ban on animal grazing and the collection of these species for medicinal purposes. Additionally, cultivating these plants in fields can support the sustainable use of their medicinal and edible properties. Propagation outside of natural habitats, such as in botanical gardens or research centers, can also be effective. Once these plants are cultivated, they can be reintroduced to strengthen wild populations. Furthermore, maintaining these species in gene banks is vital for conservation under extreme conditions. The preserved seeds or plant parts can assist in future habitat restoration or be cultivated in botanical gardens if natural conditions become unsuitable. We have also added these to the manuscript text.

---

## [Editor Report · Decision Letter 1]

3 Mar 2025

Assessing the impact of global warming on the distributions of Allium stipitatum and Kelussia odoratissima in the Central Zagros using a MaxEnt model

PONE-D-24-55210R1

Dear Dr. Khajoei Nasab,

We’re pleased to inform you that your manuscript has been judged scientifically suitable for publication and will be formally accepted for publication once it meets all outstanding technical requirements.

Kind regards,

Sara Hemati

Academic Editor

PLOS ONE

Additional Editor Comments (optional):

Accept
---

## [Editor Report · Acceptance letter]

PONE-D-24-55210R1

PLOS ONE

Dear Dr. Khajoei Nasab,

I'm pleased to inform you that your manuscript has been deemed suitable for publication in PLOS ONE. Congratulations! Your manuscript is now being handed over to our production team.

Kind regards,

on behalf of

Dr. Sara Hemati

Academic Editor

PLOS ONE